# The Mining and Technology Industries as Catalysts for Sustainable Energy Development

**Katundu Imasiku** [1,*] and **Valerie M. Thomas** [2,3]

1   African Center of Excellence in Energy for Sustainable Development, University of Rwanda, Kigali 4285, Rwanda
2   School of Industrial and Systems Engineering, Georgia Institute of Technology, Atlanta, GA 30322, USA; valerie.thomas@isye.gatech.edu
3   School of Public Policy, Georgia Institute of Technology, Atlanta, GA 30322, USA
*   Correspondence: katunduimasiku@gmail.com

**Abstract:** The potential for mining companies to contribute to sustainable energy development is characterized in terms of opportunities for energy efficiency and support of electricity access in mining-intensive developing countries. Through a case study of the Central African Copperbelt countries of Zambia and the Democratic Republic of Congo, energy efficiency opportunities in copper operations and environmental impact of metal extraction are evaluated qualitatively, characterized, and quantified using principles of industrial ecology, life cycle assessment, and engineering economics. In these countries the mining sector is the greatest consumer of electricity, accounting for about 53.6% in the region. Energy efficiency improvements in the refinery processes is shown to have a factor of two improvement potential. Further, four strategies are identified by which the mining and technology industries can enhance sustainable electricity generation capacity: energy efficiency; use of solar and other renewable resources; share expertise from the mining and technology industries within the region; and take advantage of the abundant cobalt and other raw materials to initiate value-added manufacturing.

**Keywords:** copper; cobalt; mining; sustainable energy development; engineering economics; multi-national enterprises

## 1. Introduction

Many of the world's poorest countries rely on mining and fossil fuel extraction as the primary basis of their economy [1]. Development pathways for resource-intensive economies are a core challenge of sustainable development [2]. Despite the great wealth of the materials mined and exported, leveraging these industries into economic benefits for the people has been a continuing challenge. Less resource-intensive countries, together with all the world's rich and middle-income countries, have developed through a transition to a manufacturing economy using fossil fuels countries [3]. However, the manufacturing pathway for developing economies might not be realistic for all countries [4]. A key part of this new development model is vested in the role of multi-national enterprises in supporting sustainable economic development in these regions.

The electronics industry depends on minerals from a number of the foremost economically challenged areas of the world. The industry has in recent years taken responsibility for its supply chains in areas characterized by conflict, child labor, and widespread environmental impacts from mining. Some of the world's largest mining firms have developed a framework for supporting sustainable development in its operations and supply chains [5].

To date, the key topics addressed by multinational sustainable development efforts have been child labor, conflict, the rights of indigenous people, human rights, fair trade, and the environmental impacts of mining, extraction and refining.

This study introduces the concept that mining multi-national enterprises (MNEs) can improve their operations and contribute to energy systems for sustainable development. Energy and its accessibility are at the core of social, economic and environmental concerns facing all nations, especially in developing countries. In several countries, mining and ore refining is the largest electricity consumer. The study explores the potential role of multinationals in energy development through two case studies: Zambia and the Democratic Republic of Congo (DRC), which is also referred to as Central African Copperbelt (CAC). This region is one of the world's richest mineral ore body. It is the world's second largest copper cathode producer and the world's largest cobalt producing region [6,7]. In Zambia, copper and cobalt are the major exports, accounting for more than 70% of foreign exchange earnings. This study explores options for increasing energy efficiency, renewable energy deployment, and providing technical support for energy system reliability and development.

A number of previous studies have addressed the relationship between development and environmental impacts in broad statistics analyses. Aldieri et al. found that there is a spillover effect, that is, learning from proximity to technologically advanced firms, that can support technical efficiency, economic viability, and higher productivity [8]. A review across developed and developing countries by Wang et al. finds that the studies on developed countries predominantly take a multi-industry perspective, while studies from developing countries have focused primarily on manufacturing [9].

There have been some studies on the mining industry in developing countries. Lane's study on the need for collaborations to achieve sustainable mining recognized that for the mining sector to remain sustainable, the concept of inclusive growth needs to be adopted by all players. However, the mining sector in Africa comes with challenges like social equity, social license to operate, local supplier development, new investment model development, the requirement for mines to adopt initiatives for infrastructure and energy development [10]. Lebre et al. identify the co-occurrence of environmental, social, and governance risk factors in mining for energy-relevant metals [2].

This research study evaluates the potential for improved energy systems and energy management in mining to support sustainable development. In contrast to previous work, this study quantifies energy efficiency potential for the copper industry in Zambia and DRC, and suggests additional technical measures for increased use of renewables and sharing of technical expertise to support electrification and development. Based in the field of industrial ecology [11], this work begins to fill the literature gap on quantitative assessments of how the multi-national mining and technology industries can contribute to clean energy access in developing countries.

The field of industrial ecology takes the perspective that industry can be an agent of change in meeting environmental objectives, alongside worker safety and customer satisfaction [11]. Taking a largely engineering and technical perspective, the emphasis is on identifying opportunities for meeting broad societal and environmental objectives through innovation and efficiency. Within industrial ecology, life cycle assessment is an important method for evaluating environmental impacts throughout the entire supply chain of a product or service. This is combined with engineering economic or techno-economic assessment, which evaluates the costs and benefits of different technology or policy options.

The study qualitatively characterizes and quantifies the need for energy efficiency in mine production and shows the potential for mining and technology companies to contribute to the sustainable energy development using principles of industrial ecology, life cycle assessment and engineering economics.

Although framed in an industrial ecology context, the full challenge of energy development must be shaped by local choices and decisions by citizens and governments. By developing an industrial ecology framework for the energy development challenge, we aim to shape the potential for

multi-nationals to participate as partners with governments, citizens, agencies, and local entrepreneurs in meeting sustainable development goals.

## 2. Methods

To achieve this aim, a fourfold method is adopted to first evaluate the current energy scenario in Zambia and DRC including modeling energy flow diagrams for the two cases and second, to benchmark with the leading global copper producer of mined copper, Chile to evaluate the energy requirement for copper production in Zambia and DRC. This will help to quantify the copper production efficiencies for the two main copper production methods in use, pyrometallurgical and solvent extraction (SX) and electrowinning (EW). Thirdly, the study evaluates the environmental impact of metal extraction in terms of greenhouse gas (GHG) emissions per ton of copper production while benchmarking with other top copper producers like Australia and the expected global average GHG emission quantity. Fourthly, the sharing of expertise from mining and technology industries for energy development was explored because these tech industry experts can serve on advisory panels for policy formulation and provide a knowledge base for possible replication within Sub-Saharan Africa.

Figure 1 gives a visual perspective of the adopted research method to demonstrate how the mine technology industry can be used to support sustainable energy development while fostering economic growth and development in SSA.

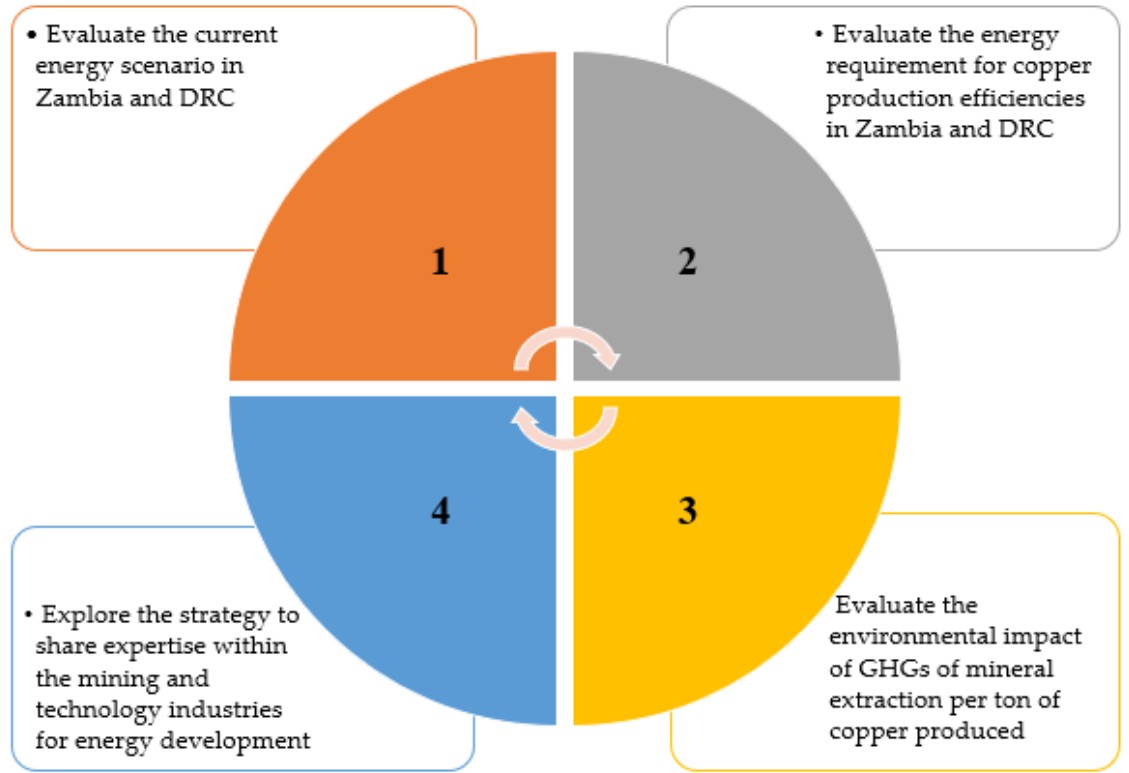

**Figure 1.** Evaluation steps for mine technology industry support for sustainable energy development.

First, the Central African Copperbelt is a mineral-rich region, but each location has a different political governance system, energy management systems and challenges which may need a different approach and methodology. However, the mining industry in CAC has synergies like having some mine MNEs operating in both Zambia and the DRC as mine owners. This background coined the evaluation of the energy scenarios in DRC and Zambia separately using energy flow diagrams (Sankey) to show the produced energy, its conversion or transformation visually and quantitatively

including primary energy, imports, exports and losses and energy consumption by economic sectors in gigawatt-hours (GWh).

Second, the energy requirement for copper production efficiencies in Zambia and DRC was then evaluated to while benchmarking the engineering economic analysis copper production per ton for Zambia and DRC with Chile, the leading global producer of mined copper. The total annual smelting capacities, measured in tons per annum, was also estimated in CAC.

Third, the environmental impact of GHGs of mineral extraction per ton of copper produced was analyzed while benchmarking with another top copper producer, Australia to qualitatively analyze the greenhouse gas emission contribution in the mining and milling sector and smelting sector according to the energy resource used in the different mining activity or process.

Finally, the need to share this expertise for broader energy development projects could make a significant impact on the energy and economic development in DRC and Zambia. Apart from benchmarking with Chile, which has already deployed solar energy on a large scale as a main source of electricity for copper production, this stage explores the strategy to share expertise within the mining and technology industries for energy development.

## 3. Results

### 3.1. Energy in Zambia and the Democratic Republic of Congo

Zambia's electricity generation is dominated by hydropower that comprises over 95% of total generation capacity. About 90% of hydro generation comes from just two power stations, the Kariba North Bank and the Kafue Gorge, in the country's Southern province. The country additionally generates about 0.5 GWh using diesel generation system for standby supply. Zambia's sole focus on hydroelectricity is understandable but makes it vulnerable to drought. Zambian power plants are summarized in Table A1 (Appendix A).

Zambia has an average national electrification rate of about 31%, with 61% of the urban areas having access to electricity, while the rural community electrification rate is only 4% [12,13]. Figure 2, of the energy flow for electricity in Zambia, shows that more than 52.3% of the electricity is used by industry, almost all of which is for the metals industry. Zambia's key mine MNEs are Glencore, Mopani, Vedanta Resources, Konkola Copper, First Quantum Minerals, Kalumbila, Barrick Gold and Lumwana mines [14].

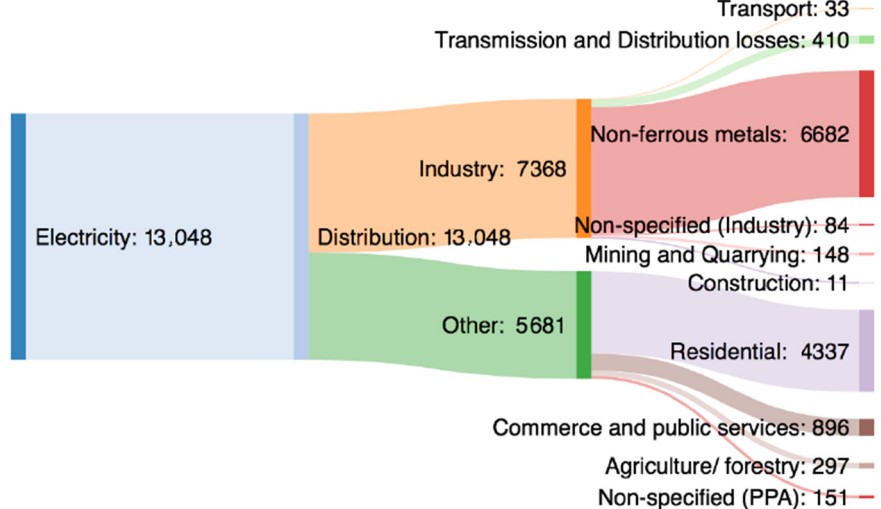

**Figure 2.** Electricity energy flow in Zambia, as of 2018 (GWh), as built using SankeyMATIC.

The DRC is endowed with large mineral resources and has the potential to install up to 100,000 MW of hydropower capacity. The installed capacity is 2790 MW. Almost 8% of the produced hydropower is

consumed by the residential sector while other sectors mostly consume oil. Only 1% of the people in rural areas of DRC and about 8% in the urban areas have access to hydroelectric power [15,16]. The country's electrification rate remains low at 9.6%, and the government's vision is to increase the level of service up to 32% in 2030 [17].

The major mining activities in DRC are in Katanga province whose power capacity is around 900 megawatts (MW) but only about 461.7 MW is available for usage, due to several issues including poor power governance and lack of support from government; lack of findings with high electricity tariffs that are not cost reflective; poor electric utility performance characterized with poor recovery of public consumption electricity invoices that are as high as 40% of total consumption; absence of a regulatory agency and a Rural Electrification Agency; high taxes, and import duties and most importantly the current installed hydropower system has poor infrastructure that leads to many technical losses in the transmission and distribution networks of SNEL. This has resulted in a low distribution efficiency of hydropower with a far lower distributed power compared to the installed power (Figure 3).

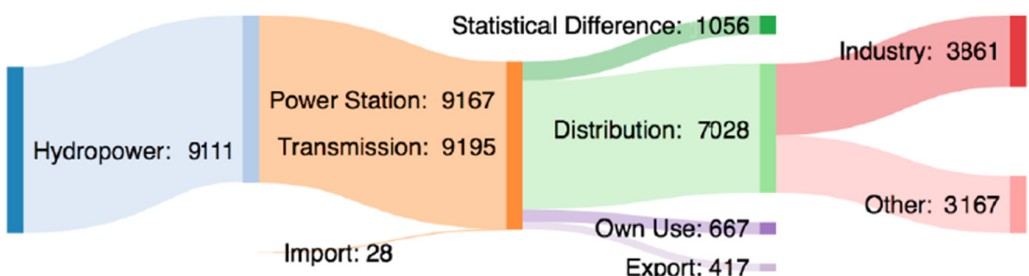

**Figure 3.** Electricity energy flow in the Democratic Republic of Congo (DRC), as of 2016 (GWh), as built using SankeyMATIC.

The low generation cost of DRC's largest power plant, Inga, potentially reduces the overall cost of electricity in DRC, making it cheap and affordable, but this is subject to other factors like good governing policies and political stability [18,19]. Table A2 (Appendix A) shows the installed, planned electricity capacity under development in DRC as of 2018.

In the past, DRC had huge losses in transmission and distribution; therefore, only 1228 MW (almost 50% of installed capacity) was available capacity in DRC [19]. This situation improved subsequently by 2018 [20]. Figure 3, of the electricity flow in the Democratic Republic of Congo, shows that more than half (54.9%) of the electricity is consumed by the industry sector, which is predominantly mining. The statistical differences in Figure 3 account for transmission losses while rationale of the Other sector comprises 2444.44 GWh for the Residential sector and 722.22 GWh for Commerce and Public Services sector.

### 3.2. Copper and Cobalt Production Efficiency Potential

Copper production in Zambia and the DRC is benchmarked with Chile, the leading global producer of mined copper. Table 1 shows the energy use for copper production in Chile, Zambia, and the Democratic Republic of Congo. The two main methods used in copper production are pyrometallurgical and solvent extraction (SX) and electrowinning (EW).

**Table 1.** Energy requirement for copper production in Chile, Zambia and DRC.

| | Copper Production (Mt/year) | Electricity for Cu Production (TWh/year) | Production Method: Concentrate (%) | Production Method: SX-EW (%) | Energy Efficiency (kWh/kg Cu) |
|---|---|---|---|---|---|
| Chile | 5.6 | 22 | 63 | 33 | 3.9 |
| Zambia | 0.755 | 6.2 | 57 | 43 | 8.2 |
| DRC | 0.85 | 7 | 90 | 75 | 8.2 |

Energy consumption in copper mining and refining varies with the nature of the ore. Copper production operations in DRC and Zambia has the advantage of high ore grades (3 to 5% acid-soluble Cu), and consequently significantly higher pregnant leach solutions (PLS) copper grades found in other regions [21].

The two main copper extraction processes are the pyrometallurgical process and the solvent extraction and electrowinning (SXEW). The pyrometallurgical processing involves crushing of abundant sulfide ore deposits that are mined and then concentrated into copper concentrate, containing around 30% pure copper which is then smelted into copper anode of 99.5% purity. The copper concentrate utilized in refineries to produce copper blister, which contains about 98% pure copper. The blister is further processed to produce pure copper cathode at copper refineries through leaching, SX and EW to make copper cathodes. The SXEW process is energy intensive (~2.1 MWh/t Cu) because it directly produces copper cathodes from oxide ores. The pyro-metallurgical process is used to recover copper from the mined ore, while SXEW plants process mostly oxide ores but not scrap [22].

The annual smelting capacities are: 870,000 tons per annum at Mufulira smelter, 311,000 tons per annum at Nchanga smelter, 300,000 tons per annum at Nkana Refinery, 150,000 tones of blister copper per annum at Chambishi smelter and 25,000 tons of blister copper at Chambishi Metals refineries and 336,000 tons at Kansanshi smelter [23,24].

The Kansanshi Mine Smelter, identified as the most advanced copper smelter in the world, is worth $900 million and started operating in early 2015 in North-Western province of Zambia. Owned by First Quantum Minerals' (FQM) the Kansanshi Mine Smelter is of strategic importance to both DRC and Zambia. First, the significance is because Zambia's North-Western province has three copper mines which produce more than half of Zambia's copper. Second, since Kansanshi Mines in Zambia and Sentinel copper mine in DRC are both owned by First Quantum Minerals (FQM), the copper concentrate from Sentinel copper mine in nearby Kalumbila in DRC are refined at the new refinery in Zambia where acid is abundant to treat oxide copper ore during the refinery process [25].

More broadly, due to robust domestic smelting capacity, rising mine output and continued support from the Zambian government, the smelting capacity has also improved due to large investment by multi-nationals, making Zambia the largest refined copper producing country in Africa, ahead of DRC and South Africa. The Zambian copper mines are forecast to continue to grow.

The three main competitive refinery technologies are reverberatory furnaces, flash furnaces and heap leach-SX-EW. These three technologies can be compared on the basis of ore grade, scale and other features. Out of these three technologies the heap leach-SX-EW is more modern although it is still being improved. Previously the SX-EW process had an extractant consumption of about 3 kg per ton of copper but this has been improved to about 2 kg per ton of copper. Reverberatory furnaces and flash furnaces can achieve a total recovery of 88%, whereas the SX-EW method achieves a marginal 62%. On the contrary, concerning energy consumption, the flash smelters consume about 6000 kWh/t, and the reverberatory smelters consumes 5700 kWh/t, whereas the SX-EW smelters can consume as much as 9350 kWh/t [26]. The engineering economic analysis to evaluate the energy requirement for copper production is given in Table 1.

Table 1 indicates that copper production in Zambia and DRC could potentially increase its energy efficiency by a factor of two (50%) [27]. A more detailed local analysis would be needed to fully characterize the energy efficiency potential. An interview with a Zambian mineral processing specialist Dr. Kenneth Sichone, on 5 September 2018, confirmed that some MNEs that invested in Zambia in early 2000, after the privatizations of mines, used Chilean processing technology, without considering the concentrate types and copper ore grade against the design specification of the processing technologies and this reduced the production levels [28]. Globally, average production efficiencies can be as low as 60 GJ/t (8.34 kWh/kg) of copper due to low ore grades, ranging to about 30 GJ/t (4.2 kWh/kg) of copper for the most efficient refinery systems. The production efficiency reduces as the ore grade of copper reduces especially to as low as 0.1% [26]. On a global basis, energy contributes approximately 35% of total copper refinery production costs [29].

*3.3. Environmental Impact of Metal Extraction*

In the Australian copper industry, the main contributor to greenhouse gas emissions are the mining and milling, accounting for about 57% while smelting accounts for about 27% [30]. The extraction of metal ores involving both underground and open-pit mining techniques have different energy consumptions and GHG emissions or waste. The selection of an appropriate processing technique depends on many factors but not restricted to the character of the ore and the location of the ore, its size, at what depth and most importantly the concentrate or ore grade. Unlike the open-pit mining method, underground mining consumes a lot of energy owing to larger demands from hauling, ventilation and water pumping. In Australia, most of the copper ore comes from underground mines [31]. The global average of greenhouse gas (GHG) emissions was estimated in 2012 to be 6.58 t $CO_2$ per ton of copper production [32].

The $CO_2$ emissions of copper processing at different refinery stages depend on the energy source used in that stage which could be coal, diesel, natural gas and solar, hydropower or any other renewables energy resource. As ore grade decreases, higher energy consumption is expected for future copper production [32].

Further, to enhance competitiveness by resolving societal problems firms should be credited at country and regional level especially if benchmarked against the international mining policies that govern the regional mining policies [33].

*3.4. Expertise from Mining and Technology Industries for Energy Development*

In Zambia and the Democratic Republic of Congo (DRC), hydropower is the main source of electricity. As discussed above this contributes to the production of copper and cobalt production with relatively low greenhouse gas emissions.

In Chile, solar power is becoming the main source of electricity for copper production [34]. The Chilean mining region has outstanding solar resources and few other energy resources, making the location particularly well suited for solar power development. However, given the success of Chilean use of solar power in the metals industry, there may be potential to introduce utility-scale solar to the Zambia and DRC mining districts.

The extensive energy consumption of mine operations can be utilized to manage the introduction of intermittent renewables such as wind and solar into the national electricity system [35]. Similar to the use of electric vehicle charging to buffer the variability of solar power [36], the operations of mining can also provide flexibility to reduce their costs and electricity system costs.

Metals and technology industries have developed substantial expertise in electricity systems operation. Sharing this expertise for broader energy development projects could make a significant impact on the energy and economic development in these countries [37].

**4. Discussion**

The technology industries use copper and cobalt for production of electronics and batteries; these are key contributors to a global sustainable energy future. The production of copper and cobalt in the Democratic Republic of Congo (DRC) and in Zambia does have some strong sustainability features, including use of renewable energy sources in the production process, and increased energy efficiency over time. Even so, at a national level the energy systems of DRC and Zambia are underdeveloped, and electricity access is low. The metals industries are the main users of electricity, and energy costs are significant for them. The situation presents an opportunity for the metals industries and the technology industries to catalyze improved energy access because energy efficient production can save about 50% of energy. This can allow both Zambia and DRC to increase the consumption of other uses to about 75% as opposed to the initial 46.4% average power to sectors other than the mines in CAC.

To design a sustainable energy supply system in CAC within the mines, both generation-mix and energy efficiency are important components. Sustainable mining can make the business more viable

because renewables can be cheaper than conventional power. Moreover, improved socio-economic development of both society and the mining firm benefits the nation. Requirements from the customers of mining companies, throughout the international supply chain, are increasingly pushing suppliers to meet sustainability standards. Achieving socio-economic development implies that renewable energy will not simply be used to solve societal problems concerning the environment but support the business model of the international mining industry and potentially provide benefit more reliable energy supply at a reduced cost. From a business perspective, this provides corporate shared value amongst all stakeholders.

The extraction of metal ores involving both underground and open-pit mining techniques have different energy consumptions and GHG emissions or waste. Further, the selection of an appropriate processing technique depends on many factors but not restricted to the character of the ore and the location of the ore, its size, at what depth, and most importantly the concentrate or ore grade. Unlike the open-pit mining method, underground mining consumes a lot of energy owing to larger demands from hauling, ventilation, and water pumping.

The $CO_2$ emissions of copper processing at different refinery stages depend on the energy source used in that stage which could be coal, diesel, natural gas and solar, hydropower, or any other renewables energy resource. A quantitative analysis of greenhouse gas emissions from cobalt and copper production in Zambia and the DRC is beyond the scope of this study. However, the dominance of hydropower in the electricity systems of both countries suggests that the greenhouse gas emissions would be lower than that produced in Australia, which has a fossil fuel-dominated electricity system. As ore grade decreases, higher energy consumption is expected for future copper production.

The metals and technology industries have developed substantial expertise in electricity systems operation. Chile uses solar power in the metals industry, and this shows that there may be potential to introduce utility-scale solar to the Zambia and DRC mining districts. The combination of solar power with hydropower can increase the overall resilience of the electricity system, mitigating the tendency of hydropower to be affected by drought and seasonal variations. Implementation of state-of-the-technology solar systems for metals development in Zambia and DRC could benefit the metals industries and could also provide a technological foundation for the broader implementation of solar power in these countries and the broader region. Sharing this expertise for broader energy development projects could make a significant impact on the energy and economic development in these countries. Further, the energy ministries and utility systems of these countries could consider inviting the metals industry and tech industry experts to serve on advisory panels and to find ways to support the beneficial development of energy access. Situating the sharing of expertise in the concept of United Nations or other international development agency activities can provide a framework for sharing of information and access to innovative expertise in a way that is transparent and that could be replicated throughout Sub-Saharan Africa and elsewhere.

Other innovative spring-out opportunities that would benefit the local people in DRC and Zambia concern the use of cobalt in lithium-ion batteries and renewable energy technologies, power grid stabilization, and electric vehicles. The mine MNEs in Zambia and DRC export unprocessed copper and cobalt and purchase back finished battery components in CAC to support the energy storage industry. Exploring value-added manufacturing of battery components using readily available raw materials (cobalt and copper) would support sustainable development in CAC.

Multi-national companies in the metals and technology sectors have strong sustainability and social responsibility programs that support the communities and environments in which they operate. Some multi-national enterprises have set high employment standards in compliance with industry best practice and social responsibilities as stipulated by the International Council on Mining and Minerals [7]. Some multi-national firms that source materials from the CAC, notably including Apple, have worked towards reduced environmental impacts and seek to be trendsetters for other multi-nationals [38]. In 2018, Transparency International (TI) worked to ensure that multi-national companies, such as Apple, meet the appropriate labor standards concerning the procurement of cobalt

from firms that observe sustainable mining practices and not from child miners in DRC [39,40]. In the energy area, in 2018 Apple declared its complete migration to 100% renewable energy and encouraged other multi-nationals to emulate it [41].

Including a focus on energy development in the countries that supply essential materials, as part of these social responsibility programs, could help transform the energy policies and energy systems of these mining-dependent nations and provide an example of the potential for extractive industries to make positive substantial contributions to development.

## 5. Conclusions

Mining activities absorb much of the electricity generated in Zambia and the DRC. Copper production efficiency in the CAC is relatively low. With the high ore grades averaging in CAC, the energy efficiency in the refinery processes can be improved by about a factor of two. As this electricity-generating capacity already exists, reducing the energy demand would improve electricity availability in Zambia and DRC at a low cost. The implementation of more energy efficient production would benefit the mining industries by reducing production costs, and would benefit the region as a whole. The key finding, that energy efficiency measures in one industry could substantially improve energy access in two nations, suggests a new avenue to meeting sustainable energy development goals. In Zambia and the DRC, and potentially in other countries, a major portion of future energy needs can be met through energy efficiency rather than entirely by constructing new generating sources.

The mining and technology industries can enhance sustainable electricity generation capacity by deploying energy efficiency strategies throughout the mine operations. They can identify the areas or mine operation activities for renewable resource usage especially the abundant solar energy resource. The intermittency of solar energy could potentially be mitigated with battery energy storage Use of cobalt-containing storage batteries would provide Zambia and DRC with the technological benefits of their mining industries. The mining MNEs can enhance the energy storage systems by taking advantage of the abundant Cobalt raw material to explore battery component manufacturing to boost economic diversity beyond Copper and Cobalt mining in the CAC region. Further, the sharing of expertise from the mining and technology industries within the region can also support sustainable energy development and mining.

The $CO_2$ emissions of copper processing at different refinery stages depend on the energy source used in that stage which could be coal, diesel, natural gas and solar, hydropower, or any other renewables energy resource. Renewable energy resource generation-mix strategy is encouraged to achieve, net-zero emission energy generation systems within the copper mines in CAC.

This study has used a case study approach to examine the potential for technology and mining companies to contribute to energy development while meeting their own environmental goals. Expanding on this perspective, a comprehensive analysis of the potential for energy efficiency and renewable energy in mining to support both developing county energy goals and technology firms' sustainability goals could provide a basis for concerted programs in clean energy access. In addition, a more detailed quantitative industrial ecology study in the CAC region could quantify energy flows through mine industrial systems and quantify the greenhouse gas emission contributions within the mine operation processes for strategic energy modeling and optimization using a life cycle assessment. This would support future mine operation designs to increase renewable energy applications and exploit the abundant renewable resources as a leapfrog strategy for the MNEs to achieve sustainable energy development.

**Author Contributions:** Data curation: K.I. and V.M.T.; writing-original draft preparation: K.I.; writing—review and editing: K.I. and V.M.T.; visualization: V.M.T.; supervision formal analysis: V.M.T.; funding acquisition: K.I.; investigation: V.M.T. and K.I.; methodology: K.I.; project administration: V.M.T.; resources: K.I. All authors have read and agreed to the published version of the manuscript.

**Funding:** This research received no external funding.

**Acknowledgments:** The authors would like to thank the African Center of Excellence in Energy for Sustainable Development and the University of Rwanda for their support.

**Conflicts of Interest:** The authors declare no conflict of interest.

## Appendix A

**Table A1.** Installed Generation Capacity in Zambia as of 2018 (Energy Regulation Board of Zambia, 2018).

| Power Station | Installed Capacity (GWh) | Generation Type | Operator |
|---|---|---|---|
| Kafue Gorge | 5142.56 | Hydro | ZESCO |
| Kariba North Bank | 3740.04 | Hydro | ZESCO |
| Kariba North Bank Extension | 1870.02 | Hydro | ZESCO |
| Victoria Falls | 561.01 | Hydro | ZESCO |
| Lunsemfwa and Mulungushi | 290.89 | Hydro | LHPC |
| Maamba | 1558.35 | Coal | Maamba Collieries |
| Ndola | 571.40 | Heavy Fuels | Ndola Energy Plant |
| IPP Small Hydro ** | 627.24 | Hydro | IPPs |
| Small Hydro * | 225.96 | Hydro | ZESCO |
| Isolated Generation *** | 18.7 | Diesel | ZESCO |
| CEC (Stand By) | 415.56 | Diesel | CEC |
| (Solar Samfya) (Sinda, Kitwe Samfya) | 5.66 | Solar | CEC, Muhanya and Rural Electrification Authority |
| Total Installed Capacity | 15,027.39 | | |

Note: * Lusiwasi, Musonda falls, Shiwang'andu, Chishimba falls and Lunzua; ** Zengamina Power Limited and Itezhi-Tezhi; *** Shangombo and Luangwa.

**Table A2.** Installed generation capacity in DRC by 2018.

| Power Station | Capacity (MW) | Generation Type | Status | Name of River | Source |
|---|---|---|---|---|---|
| Inga I | 351 | Run of river | Operational | Congo | [42] |
| Inga II | 1424 | Run of river | Operational | Congo | [42] |
| Nseke | 260 | Reservoir | Operational | Lualaba | [19,43] |
| Ruzizi I | 40 | Reservoir | Operational | Ruzizi | [19] |
| Ruzizi II | 45 | Reservoir | Operational | Ruzizi | [19] |
| Ruzizi III | 147 | Run of river | Under Construction | Ruzizi | [19] |
| Ruzizi IV | 200 | Run of river | Proposed | Ruzizi | [44] |
| Rutshuru | 13.8 | Run of river | Operational | Rutshuru | [45] |
| Mutwanga | 10 | Run of river | Operational | Semliki | [46] |
| Nzilo | 108 | Run of river | Operational | Lualaba | [19,47] |
| Mobayi | 11.5 | Reservoir | Operational | Ubangi | [48] |
| Koni | 36 | Reservoir | Operational | Lufira | [47] |
| Mwadingusha | 71 | Reservoir | Operational | Lufira | [49] |
| Kyimbi | 18 | Reservoir/Waterfall | Operational | Kyimbi | [44] |
| Tshopo | 19.65 | Reservoir/Waterfall | Operational | Tshopo | [19,48,50] |
| Kilubi | 9.9 | | Operational | | [44] |
| Lungundi | 1.6 | | Operational | Kasai | [44] |
| Mpozo | 2.21 | | Operational | Mpozo | [44] |
| Sanga | 12 | Reservoir/Waterfall | Operational | Inkisi | [51] |
| Zongo | 75 | Reservoir | Operational | Insiki | [19,47] |
| Zongo II | 150 | Reservoir | Operational | Insiki | [52] |
| Katende | 64 | Run of river | Under Construction | Daugava | [53] |
| Kakobola | 10.5 | Run of river | Under Construction | Rufuku | [54] |
| Grand Inga | 39,000 | Run of river | Proposed | Congo | [42] |
| Inga III | 4800 | Run of river | Under Construction | Congo | [42] |
| IPPs owned by 14 Mine MNEs | 100.3 | Hydro-electric | Operational | Various Mine sites | [44] |
| Other 25 Thermal Plant IPPs | 31.6 | Thermal Power Stations | Operational | Various sites | [44] |
| Planned Capacity | 39,000.00 | | | | |
| Under Construction | 5021.50 | | | | |
| Total Installed | 2790.56 | | | | |

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
