# Peer review of "The Mining and Technology Industries as Catalysts for Sustainable Energy Development"

_sustainability, doi:10.3390/su122410410_

Round 1

Reviewer 1 Report

In abstract, do not use abbreviations. The article should be written in an impersonal form. Don't use the form "we".

Abstract is not written correctly. I suggest authors provide the background, methodology, main results. The current structure is incorrect. In line 31-32, there are some statements that would be in other literature: https://doi.org/10.3390/en13081925, https://www.sciencedirect.com/science/article/abs/pii/S0921344913002127

In introduction, please review the previous studies. For instance, are there studies similar to this study out there? If so what are they, and how does this research relate to extant literature? Or is this one of the very first studies in this area? It is worth to mention that the authors do not include any kind of literature review, this is an important weakness for an academic paper. What is the novelty for this article compared with existing studies? What is the novelty of this article? In section 1, the research problem and purpose of this study were not described clearly. This remark was made in the previous round of reviews - the authors did not correct it.  The test method is not well described. The description shows nothing, it's just a few sentences, not a description of the method. Section 3 contains only the test results. There is no discussion of the research results in this chapter. Presenting the results means nothing, if you do not discuss them in detail. Finally, they do not point to any of the clear limitations that this study shows.

There are no conclusions at work. Section 4 is a summary, not a conclusion.

For all that has been discussed before, I would recommend to the authors further development of their present work.

Author Response

We thank the reviewer for the comments for us to improve our work. In response we have fully revised the manuscript and further developed our work. The revised manuscript is more concise, articulate, and enhanced with reference to recent studies concerning the role of technology in the resources sector in developing nations and developed nations. We have added a new discussion section and a new conclusion section to fully discuss the results with enhanced conclusions and improve the flow of the paper. Although we acknowledge we may not have not fully met all the suggestions of the reviewers, this is now a much stronger manuscript as a result of the reviews and our work to meet the expected level of analysis.

Reviewer 2 Report

The article has five parts:

  • Introduction (1 page)
  • Methods (1/2 page)
  • Results and Discussion (4 and 1/2 pagaes)
  • Conclusions (2 pages)
  • References (3 and 1/2 pages)

The results and discussions are longer than the article itself? And the article itself is only half page? References are longer than the article?

There is something wrong. 

Author Response

We thank the reviewer for your comments for us to improve our work. In response we have fully revised the manuscript and further developed our work. The revised manuscript is more concise, articulate, and enhanced with reference to recent studies concerning the role of technology in the resources sector in developing nations and developed nations. We have added a new discussion section and a new conclusion section to fully discuss the results with enhanced conclusions and improve the flow of the paper. Although we acknowledge we may not have not fully met all the suggestions of the reviewers, this is now a much stronger manuscript as a result of the reviews and our work to meet the expected level of analysis.

Reviewer 3 Report

Comments and Suggestions for Authors

Overall impression:

The manuscript is certainly interesting, studying a key economic activity for two central African countries. However, several comments and changes are needed before the manuscript could be considered to be published.

Major issues.

  1. The aim of this research is to wide and vague, … “to evaluate the mining technologies in Zambia and

DRC concerning hydro-dependency syndrome to improve energy security with solar-hydro generation-mix strategy while leveraging on energy efficiency and net-zero carbon technologies to reduce the GHGs and the associated environmental”. How come?  In which way? Any evaluation is presented. Please explain, must important the conclusion sections did not stablish clearly in which fashion this goal was accomplished

  1. The research goal must be place at the end of the introduction, not in the methods section.

  1. Usually, it is not proper the use the term We as researchers, this must be neutral, this research, etc. Choose the proper tense language and expressions for each section.

  1. Methods section:

a fourfold method is adopted to first evaluate the current energy scenario in Zambia and DRC including modeling energy flow diagrams for the two cases and second, to benchmark with the leading global copper producer of mined copper, Chile to evaluate the energy requirement for copper production in Zambia and DRC. This will help to quantify the copper production efficiencies for the two main copper production methods in use, pyrometallurgical and solvent extraction (SX) and electrowinning (EW). Thirdly, we evaluate the environmental impact of metal extraction in terms of greenhouse gas (GHG) emissions per ton of copper production while benchmarking with other top copper producers like Australia and the expected global average GHG emission quantity. Fourthly, we explored the sharing of expertise from mining and technology industries for energy development because these tech industry experts can serve on advisory panels for policy formulation and provide a knowledge base for possible replication within sub-Saharan Africa”.

Is a large sentence without describing what type of analysis was developed, which datasets analyzed, equations applied, etc. The section looks like a result of a search on references, like a continuity of the introduction section.

  1. Figure 3 and 4 sources? Please provide the sources of both Figures, are they developed by authors? Or extracted from another publication? Please cite properly.

  1. Results and discussion section, this a key section, however there is no further analysis, again the section seems as a continuity of the introduction or even a data sources section, in spite of a results and discussion, actually there are any discussion since there are not any analysis.

  1. In your goal you stablish the following:

Thirdly, we evaluate the environmental impact of metal extraction in terms of greenhouse gas (GHG) emissions per ton of copper production while benchmarking with other top copper producers like Australia and the expected global average GHG emission quantity.

However, in the results discussion section you claim:

A quantitative analysis of greenhouse gas emissions from cobalt and copper production in Zambia and the DRC is beyond the scope of this study

It seems contradictory. Then the question is, how deep was your evaluation or assessment of the quantitative analysis of greenhouse gas emissions from cobalt and copper production in Zambia and the DRC since it is beyond the scope of this study? No Table, figure, even a more detailed explanation involving data, comparing both countries is provided. Please, this part of the section needs to be carefully revised.

  1. The clarity of the presentation of the article and the results would be improved by the inclusion of graphics. In my opinion the data and results as presented are difficult to follow and interpret.

  1. Results and discussion section, there are parts in this section that do not derive from the study carried out but from contributions from other authors, seems more as a literature review results.

  1. In my opinion, the graphic representation of the detected results would help greatly for the section, it would even be convenient in my opinion to be able to graphically represent in some way the results obtained that, apart from supporting the results, would also facilitate the monitoring of the conclusions section.

Author Response

(The authors gave the same response as above.)

Reviewer 4 Report

The paper "The Mining and Technology Industries as Catalysts for Sustainable Energy Development" is interesting for journal readers but the current version of the manuscript needs revisions before further consideration.

The aim of the analysis should be evidenced in the abstract and introduction sections.

The literature review should consider the recent contributions concerning the role of technology in the resources fields for the sustainable development (Aldieri et al., 2020; Wang et al., 2019).

The methodological approach should be further explained for a full comprehension of the investigation.

The results should be further discussed also in terms of policy implications and should be compared to those of the main literature.

References.

Aldieri L., Makkonen T. and Vinci C. P. (2020). Environmental knowledge spillovers and productivity: A patent analysis for large international firms in the energy, water and land resources fields. Resources Policy, https://doi.org/10.1016/j.resourpol.2020.101877.

Wang H., Huang J., Zhou H., Deng C. and Fang C. (2019). Analysis of sustainable utilization of water resources based on the improved water resources ecological footprint model: A case study of Hubei Province, China. Journal of Environmental Management, https://doi.org/10.1016/j.jenvman.2020.110331.

Author Response

(The authors gave the same response as above.)

Round 2

Reviewer 1 Report

The authors did not make any significant changes to the introduction (first review round). Therefore the comment from the 1st round is still valid. References are given and should be added.

Need to strengthen these points:
1. Clarity on the urge and novelty in the introduction section.
2. Concrete literature gap.
3. Arguments developed with less literature concern. Needs to back the considered problem with relevant theories.
4. Discussion needs to developed in the sense of bringing new objectives or contexts to explore in the future.
5. Need strong comment on scientific outcomes

Author Response

We thank the reviewer for the comments and support. We have revised the manuscript and we discuss our responses in the attached document.

Reviewer 3 Report

Thanks for your effort to improve the manuscript, looks and actually is a better version. However, I still have some question regarding the whole manuscript, hopefully you can consider this to improve the final version before been published.

Since Hydro energy is proved to be one of the environmentally friendly forms and reliable mostly to high consumption industrial activities such is mining metals, then what is the research is proposing alternate sources of energy such is solar or another renewable sources.

Also, as it was mentioned in section 3.1, I am questioning if the goal is to improve energy efficiency to help increase the national (or rural) electrification rates in both countries? It is an attempt from the industry taken responsibility for its supply chains in these two countries?

Author Response

(The authors gave the same response as above.)

Reviewer 4 Report

The paper has been structurally improved according to the reviewers' comments. Now, the current version of the manuscript can be accepted for the publication.

Author Response

(The authors gave the same response as above.)
